

# Impact of nitrogen application and crop stage on epiphytic microbial communities on silage maize leaf surfaces

Dan Wu*, Xueling Ma*, Yuanyan Meng, Rongjin Cai, Xiaolong Zhang, Li Liu, Lianping Deng, Changjing Chen, Fang Wang, Qingbiao Xu, Bin He, Mingzhu He, Rensheng Hu, Jinjing Zhen, Yan Han, Shaoshen He and Liuxing Xu

College of Agronomy and Life Sciences, Zhaotong University, Zhaotong, Yunnan, China
* These authors contributed equally to this work.

## ABSTRACT

This study aimed to examine the impact of nitrogen (N) fertilization on phyllosphere microorganisms in silage maize (*Zea mays*) to enhance the production of high-quality silage. The effects of different N application rates (160, 240, and 320 kg ha$^{-1}$) and maturity stages (flowering and dough stages) on microbial diversity, abundance and physiochemical properties of the leaf surfaces were evaluated in a field experiment. The results showed that N application rates did not significantly impact the abundance of lactic acid bacteria (LAB), aerobic bacteria (AB), yeasts, or molds on the leaf surfaces. However, these microbes were more abundant during the flowering stage compared to the dough stage. Furthermore, the N application rate had no significant impact on inorganic phosphorus, soluble sugar, free amino acids, total phenolic content, and soluble protein concentrations, or pH levels on the leaf surfaces. Notably, these chemical indices were lower during the dough stage. The abundance of *Pantoea* decreased with higher N application rates, while that of other microorganisms did not changes significantly. The abundance of AB, LAB, yeasts, and molds were positively correlated with soluble sugar, soluble protein, inorganic phosphorus, free amino acids, and total phenolic concentrations on leaf surfaces. Moreover, water loss was negatively correlated with the abundance of AB, LAB, yeasts, and molds, whereas water retention capacity and stomatal density were positively correlated with microbial abundance. We recommend applying an optimal N rate of 160 kg ha$^{-1}$ to silage maize and harvesting at the flowering stage is recommended.

# INTRODUCTION

Maize (*Zea mays*) silage is a widely used form of roughage due to its high yield, nutritional value, and digestibility (*Zhao et al., 2022*). In countries like United States and China, millions of hectares are dedicated to silage maize production to meet the demand for animal feed (*Bernard & Tao, 2020*). Additionally, more than 1.67 million hectares of silage maize were planted in China in 2020 (*Zhao et al., 2022*). To ensure an adequate supply of

Corresponding author
Liuxing Xu, 331405719@qq.com

feed in different seasons and climatic conditions, fresh maize is often made into silage, which effectively preserves the nutritional value of the maize and improves its palatability to ruminants (*Benchaar et al., 2014*). Various factors, including water-soluble carbohydrate concentration, buffering energy, and the presence of epiphytic plant microbes, directly influence the fermentation process of silage. Field management practices (*e.g.*, agronomic practices and harvest timing) have been studied with the aim of improving the quality of silage fermentation (*Li et al., 2016*; *Xu et al., 2021*). During ensiling, microorganisms (*e.g.*, lactic acid bacteria (LAB), butyrate bacteria, and spoilage bacteria) play a crucial role in the silage fermentation quality. However, their distribution on grass surfaces is often limited due to structural (*Vorholt, 2012*) and nutritional (*Tang et al., 2022*) factors.

Common LAB found on grass surface include *Enterococcus*, *Lactococcus*, *Lactobacillus*, *Pediococcus*, *Leuconostoc*, and *Weissella* (*Yu, Leveau & Marco, 2020*). The population sizes and clustering of LAB are often correlated with morphological characteristics of the leaves (*Tang et al., 2022*). For instance, glaucous leaves, such as those of rye (*Secale cereale*), tend to have lower microbial abundance, including lower abundance of aerobic bacteria (AB), LAB, yeasts, and molds, compared to non-glaucous leaves (*Tang et al., 2023*). Additionally, the composition of microbial communities on different crop surfaces can vary. Silage maize surfaces are mainly populated by AB species, such as *Paenibacillus*, *Flavobacteriaceae*, *Sphingomonas*, *Exiguobacterium*, *Rhizobiaceae*, *Acinetobacter*, and *Buchnera* (*Kadivar & Stapleton, 2003*). Wheat (*Triticum aestivum*) surfaces, on the other hand, are dominated by *Pantoea*, *Weissella*, *Pseudomonas*, *Exiguobacterium*, and *Paenibacillus* (*Keshri et al., 2019*). Moreover, the geographical location can also influence the microbial communities on forage leaf surfaces. For example, in southwest China, silage maize surfaces are dominated by *Weissella*, *Pseudomonas*, *Lactobacillus*, and *Leuconostoc* (*Guan et al., 2018*), while in northeast China, *Leuconostoc, Weissella, Sphingobacterium*, and *Stenotrophomonas* (*Wang et al., 2021*) are more prevalent. However, the impact of field management practices, including fertilization and irrigation, on the abundance and species composition of microorganisms on leaf surfaces remains uncertain.

Optimal nitrogen (N) fertilizer application is known to positively affect grass yield. N is very limited in soil, and agricultural systems often receive N inputs from anthropogenic sources (*Canfield, Glazer & Falkowski, 2010*), even when excessive amounts of N are not required. Recent research has explored the effects of N fertilization on microbial communities of soil (*Zhang et al., 2022*), but the potential effects of N fertilizer on number and species of microbes on grass surface are not well understood. Therefore, this study conducted a field experiment that aimed to compare the effects of different N application rates and maturity stage on the species, abundance of microbes, and physiochemical properties of silage maize leaf surfaces and on the physiochemical properties of the leaf surfaces. We hypothesized that (a) high N application rates would increase microbial diversity and abundance on silage maize leaf surfaces, (b) high N application rates would enhance nutrient levels on the leaf surfaces of silage maize, and (c) microbial diversity and abundance would be greater during the flowering stage compared to dough stage of silage maize.

## MATERIALS AND METHODS

### Experimental site

The field experiment took place in Zhaotong, Yunnan Province, China, in a region characterized by typical maize and fallow cropping systems. The specific coordinates of the site are 27°19′N and 103°45′E located at the field of Zhaotong Academy of Agricultural Sciences. The soil properties at a depth of 0–20 cm were as follows: soil organic matter 30.4 g kg$^{-1}$, total N 1.42 g kg$^{-1}$, nitrate N 307 mg kg$^{-1}$, available phosphorus 64.1 mg kg$^{-1}$, and pH 6.04. The average rainfall during the maize growing period over the past 20 years was 488 mm, and the average air temperature during the same period was 16.1 °C. The total rainfall and average air temperature during the crop growing periods were 499 mm and 22.6 °C in this study, respectively.

### Crop planting and management

Silage maize (Zhaohuang NO. 1) was sown on April 10, 2022, and harvested on October 8, 2022. The experimental treatments comprised of three N application rates treatments: 160 kg ha$^{-1}$ (low, N160), 240 kg ha$^{-1}$ (conventional, N240), and 320 kg ha$^{-1}$ (high, N320). The maturity stages evaluated were flowering and dough stages. Each test plot had an area of 20 m$^2$ (4 × 5 m) and contained five rows of maize. Each treatment was in triplicate and randomize. The seeding density was 75,000 plants ha$^{-1}$, with manual scattering of maize seeds. The N fertilizer was applied in two splits: at baseline (April 10, 2022, seedling stage) and follow-up (July 16, 2022, jointing stage), at a ratio of 6:4. No insecticides or fungicides were used during the experiment. The maize plants were irrigated four times during growth (sprinkler irrigation), with a total irrigation amount of 600 m$^3$ ha$^{-1}$. Irrigation was stopped 7 days before sampling.

### Plant materials and sample processing

For subsequent analyses, the first leaf below the ear of maize was selected from each plot at the flowering and dough stages. A sterile cotton swab was used to scrape the upper or lower surface of the leaf (4 × 6 cm), avoiding damage to the leaf veins. The collected substances were packed into sterile test tubes containing 5 mL of 8.5% NaCl solution (10 leaves were selected) to measure the abundance of AB, LAB, molds, and yeast. Additionally, ten leaves were selected for rapid transport to the laboratory in polyethylene bags under aseptic conditions. The fresh maize leaves were divided into four parts for the determination of stomatal density, moisture retention capacity, water loss, and chemical composition analysis. Leaves were also selected at the flowering stage and preserved in liquid nitrogen for microbial community analysis.

### Microbial community composition analyses

Under sterile conditions, sterile cotton swabs were used to randomly scrape off the surface microorganisms of 10 g of the leaves. Sterile water (90 ml) was added, then the mixture was shaken for 5 min on a laboratory shaker. The microorganisms were cultured on nutrient agar medium (AB, aerobic conditions at 37 °C for 1 d), and De Man, Rogosa, Sharpe (MRS) agar medium (LAB, anaerobic conditions at 37 °C for 2 d), and potato dextrose agar

(yeasts and molds, aerobic conditions at 37 °C for 3 d) were used to culture the microorganisms.

## Measurement of chemical constituents

To extract chemicals from the leaf surface 20 g of fresh maize leaf material was washed with 50 ml of distilled water (300 rpm for 5 min). The pH of the filtrate was determined using a pH meter (LE438 pH meter; Mettler Toldeo, Shanghai, China) and was then used to measure the total phenol (*Yoo et al., 2004*) and free amino acid concentrations (*Lee & Takahashi, 1966*). The soluble protein content was determined according to the method recommended by *Bradford*'s *(1976)* recommended method. Finally, the concentrations of soluble sugars and inorganic phosphorus were determined using the anthranilic-sulfuric acid (*Murphy, 1958*) and molybdenum blue (*Hande et al., 2013*) methods, respectively.

## Measurements of physiological characteristics

Leaves that exhibited optimal growth, were free of pests and diseases, unfolded, and mature (avoiding leaf veins) were selected. Stomata densities on leaf surfaces were estimated from imprints made using transparent nail varnish (*Leroy et al., 2008*). Water loss and moisture retention capacity were determined using the drying method (*Ni et al., 2015*). At the flowering stage, measurements of transpiration rate, net photosynthetic rate, intercellular carbon dioxide concentration, intercellular carbon dioxide partial pressure, pore conductivity of water vapor, total conductance of water vapor, and total conductivity of carbon dioxide of maize leaves were taken using a LI 6800 photosynthetic analyzer (LI-COR, Lincoln, NE, USA) on a sunny morning (9:00–11:00 am) with specific parameters settings including air flow rate 500 $\mu$mol s$^{-1}$, relative humidity 50%, carbon dioxide concentration 400 ppm, speed 10,000 rpm, and light intensity 1,500 $\mu$mol m$^{-2}$s$^{-1}$.

## Microbial identification

Total microbial community DNA was extracted using an EZNA®Plant DNA kit (Omega Bio-tek, Norcross, GA, USA) following to the manufacturer's instructions. DNA was extracted using 1% agarose gel electrophoresis and DNA concentration and purity were determined using a NanoDrop 2000. A nested polymerase chain reaction (PCR) assay was used to avoid problem in the sequencing analysis of plant endophytic bacteria, which arises from the similarity of chloroplast and mitochondria ribosomal RNA coding sequences to those of bacteria. The V5–V7 regions of the 16S rRNA genes of the strains were amplified with primers 779F (5′AACMGGATTAGATACCCKG-3′) and 1193R (5′-ACGTCATCCCCA CCTTCC-3′) using the thermocycler PCR system (GeneAmp 9700, ABI, Los Angeles, CA, USA). PCR products from the same sample were mixed and recovered using 2% agarose gel. An AxyPrep DNA Gel Extraction Kit (Axygen Biosciences, Union City, CA, USA) was used to purify recovered products. The recovered products were measured and quantified by a Quantus™ fluorometer (Promega, Madison, WI, USA). Purified PCR products were subjected to library construction using a NEXTFLEX Rapid DNA-Seq Kit and sequenced using Illumina's PE300 platform. Quality control and splicing of the Raw sequencing date were performed using Fastp software, and operational

**Table 1 Population sizes of microorganisms on the leaf surfaces for different treatments and maturity stage.**

| Maturity stage and treatment | | Aerobic bacteria (lg cfu g$^{-1}$ FM) | Lactic acid bacteria (lg cfu g$^{-1}$ FM) | Molds (lg cfu g$^{-1}$ FM) | Yeasts (lg cfu g$^{-1}$ FM) |
|---|---|---|---|---|---|
| N application rates (N) | N160 | 4.14 ± 0.13 | 2.72 ± 0.34 | 2.71 ± 0.16 | 3.14 ± 0.36 |
| | N240 | 4.29 ± 0.07 | 2.34 ± 0.63 | 2.65 ± 0.19 | 3.06 ± 0.33 |
| | N320 | 4.31 ± 0.08 | 2.42 ± 0.63 | 2.48 ± 0.25 | 3.23 ± 037 |
| Maturity stage (MS) | Flowering stage | 4.43 ± 0.02a | 3.58 ± 0.07a | 3.00 ± 0.09a | 3.90 ± 0.06a |
| | Dough stage | 4.06 ± 0.06b | 1.41 ± 0.29b | 2.22 ± 0.09b | 2.38 ± 0.10b |
| P | N | 0.398 | 0.877 | 0.710 | 0.948 |
| | MS | 0.000 | 0.000 | 0.000 | 0.000 |
| | N * MS | 0.118 | 0.183 | 0.526 | 0.619 |

**Note:**
Different lowercase letters in the same column represent significant difference between N application rates or maturity stage ($P < 0.05$).

taxonomic units (OTUs) were clustered and chimeric were removed based on 97% similarity using Uparse software (http://drive5.com/uparse/), based on the method of *Tkacz et al. (2020)*.

## Data processing and statistical analysis

Differences among samples were examined using one-way ANOVA and Duncan's test. Non-parametric Pearson's correlation analysis was used to determine correlations between bacterial diversity on the leaf surface and the physiochemical characteristics of the leaves. Principal component analysis (PCA) was performed on two data sets (N application rate and maturity stage) to clarify the relationships between physiochemical characteristics and microbial population sizes of the different leaves. Statistical analyses were conducted using SPSS version 19.0 for Windows (IBM Corp., New York, NY, USA) and OriginPro®2020b (OriginLab Corp., Washington, D.C., USA).

## RESULTS

### Microbial numbers

Neither the N application rate alone nor its interaction with maturity stage had any effect on the numbers of AB, LAB, yeasts, or molds ($P > 0.05$, Table 1). LAB, yeasts, and molds were less abundant on the maize leaf surfaces. At the flowering stage, the populations of AB, LAB, yeasts, and molds populations increased by 0.37, 2.17, 0.78, and 1.52 lg cfu g$^{-1}$ FM, respectively, compared to those at the dough stage ($P < 0.001$). However, the coefficient of determination ($R^2$) for the prediction of population sizes from N application rates was relatively small (Fig. 1).

### Leaf surface chemical composition

Neither the N application rate alone nor its interaction with maturity stage had a significant effect on the concentrations of leaf chemical compositions ($P > 0.05$) (Table 2). However, it is worth noting that the concentrations of inorganic phosphorus in the N240 and N320 treatments showed a reduction of 13.3% and 20.0%, respectively, compared to the N160 treatment. Similarly, the corresponding soluble sugar concentration was reduced

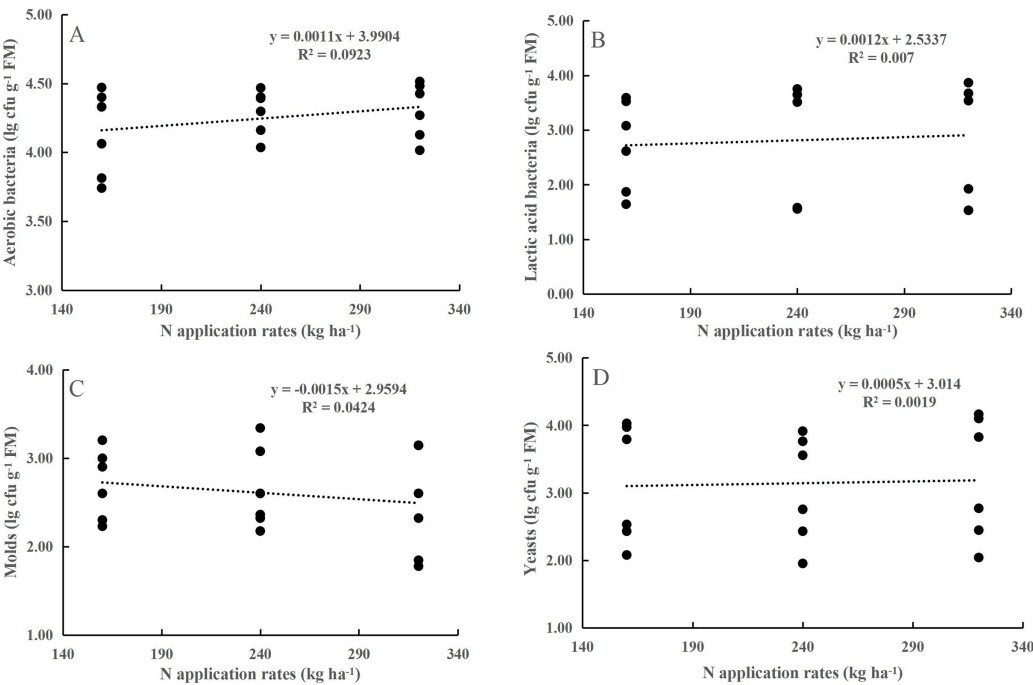

**Figure 1  Relationship between aerobic bacteria (A), yeasts (B), molds (C) and lactic acid bacteria (D) and the N application rate.**

**Table 2  Leaf chemical compositions for different treatments and maturity stage.**

| Maturity stage and treatment | | Inorganic phosphorus (ug g$^{-1}$ FW) | Soluble sugar (mg g$^{-1}$ FW) | Soluble protein (mg g$^{-1}$ FW) | Free amino acids (ug g$^{-1}$ FW) | Total phenolic (ug g$^{-1}$ FW) | pH |
|---|---|---|---|---|---|---|---|
| N application rates (N) | N160 | 60.8 ± 10.0 | 1.35 ± 0.21 | 1.01 ± 0.25 | 274 ± 105 | 434 ± 49.4 | 7.64 ± 0.04 |
| | N240 | 52.7 ± 10.9 | 1.27 ± 0.25 | 1.11 ± 0.21 | 202 ± 59.1 | 438 ± 41.2 | 7.62 ± 0.07 |
| | N320 | 48.7 ± 12.1 | 1.17 ± 0.26 | 0.86 ± 0.16 | 263 ± 82.8 | 393 ± 77.8 | 7.65 ± 0.05 |
| Maturity stage (MS) | Flowering stage | 74.8 ± 6.12a | 1.78 ± 0.04a | 1.44 ± 0.08a | 423 ± 36.2a | 543 ± 5.05a | 7.64 ± 0.06 |
| | Dough stage | 33.4 ± 3.80b | 0.74 ± 0.07b | 0.55 ± 0.04b | 69.7 ± 4.11b | 300 ± 25.3b | 7.62 ± 0.03 |
| P | N | 0.739 | 0.870 | 0.706 | 0.815 | 0.837 | 0.940 |
| | MS | 0.000 | 0.000 | 0.000 | 0.000 | 0.000 | 0.813 |
| | N * MS | 0.805 | 0.278 | 0.224 | 0.125 | 0.020 | 0.426 |

**Note:**
Different lowercase letters in the same column represent significant difference between N application rates or maturity stage ($P < 0.05$).

by 5.93% and 13.3%, respectively. Furthermore, the concentrations of leaf chemical compositions, were significantly higher at the flowering stage compared to those at the dough stage ($P < 0.001$). Specifically, at the dough stage, there was a substantial reduction in the concentrations of inorganic phosphorus (55.3%), soluble sugar (58.4%), soluble protein (61.8%), free amino acids (83.5%), and total phenolics (44.8%) reduced substantially compared to the concentrations the flowering stage.

**Table 3 Leaf physical properties for different treatments at the flowering stage.**

| Treatment | Transpiration rate (mmol $m^{-2}s^{-1}$) | Net photosynthetic rate (μmol $m^{-2}s^{-1}$) | Intercellular carbon dioxide concentration (μmol $mol^{-1}$) | Intercellular carbon dioxide partial pressure (Pa) | Pore conductivity of water vapor (mmol $m^{-2}s^{-1}$) | Total conductance of water vapor (mmol $m^{-2}s^{-1}$) | Total conductivity of carbon dioxide (mmol $m^{-2}s^{-1}$) |
|---|---|---|---|---|---|---|---|
| N160 | 4.42 ± 0.62b | 29.2 ± 0.90 | 77.8 ± 6.66 | 9.03 ± 1.05 | 146 ± 9.30b | 133 ± 13.7b | 113 ± 18.7b |
| N240 | 4.97 ± 0.24b | 31.6 ± 1.90 | 123 ± 20.3 | 10.6 ± 2.08 | 186 ± 25.6b | 179 ± 23.8b | 146 ± 19.9b |
| N320 | 7.25 ± 0.53a | 38.2 ± 4.49 | 172 ± 37.0 | 14.0 ± 3.00 | 385 ± 19.4a | 363 ± 12.8a | 220 ± 14.4a |
| P | 0.014 | 0.148 | 0.090 | 0.335 | 0.000 | 0.000 | 0.014 |

Note:
Different lowercase letters in the same column represent significant difference among N application rates ($P < 0.05$).

## Leaf physical properties

N application rates had significant effects on the transpiration rate, pore conductivity of water vapor, total conductance of water vapor, and total conductivity of carbon dioxide ($P < 0.05$). Specifically, there were no differences in these factors between the N160 and N240 treatments, but all of them were substantially lower than those in the N320 treatment (Table 3). Additionally, the moisture retention capacity at the flowering stage was higher than that at the dough stage ($P < 0.01$) (Fig. 2). A similar trend was observed for the stomatal density of maize (Fig. 3A). Moreover, the stomatal densities on the abaxial leaf surfaces in maize were higher than those on the adaxial surfaces at both the flowering and the dough stages ($P < 0.001$) (Fig. 3B). During the maturation process from flowering to dough stage, there was a gradual loss of water in the corn leaves. The water content of the leaves changed significantly 8 h ago. However, neither the N application rate nor the maturity stage had a significant effect on the rate of water loss (Fig. 2).

## Epiphytic population relationships with physicochemical traits of leaf

The abundance of AB, LAB, yeasts, and molds on the leaf surface of maize was all positively correlated with the concentrations of inorganic phosphorus, soluble sugar, free amino acids, soluble protein, and total phenolics ($P < 0.01$) (Table 4). However, the water loss rate of leaves showed a negative correlation with the abundance of these microbial species ($P < 0.05$), while the moisture retention capacity and stomatal density exhibited a positive correlation ($P < 0.01$). There was no correlation between microbial abundance and transpiration and net photosynthetic rates ($P > 0.05$).

Indicators of three N application rates and two maturity stages were plotted on a scatter diagram and analyzed using PCA (Fig. 4), where the first and second principal components explained 81.7% of the total variance in both N application rates and maturity stages. The fertility period has a significant impact on the confidence level of various indicators.

## Microbial community composition on leaves

The results of the principal coordinate analysis (PCoA) analysis showed that the different N application rates were closely grouped together, indicating high overlap of OTUs under different N application rates (Fig. 5). The leaf samples obtained from N160, N240, and

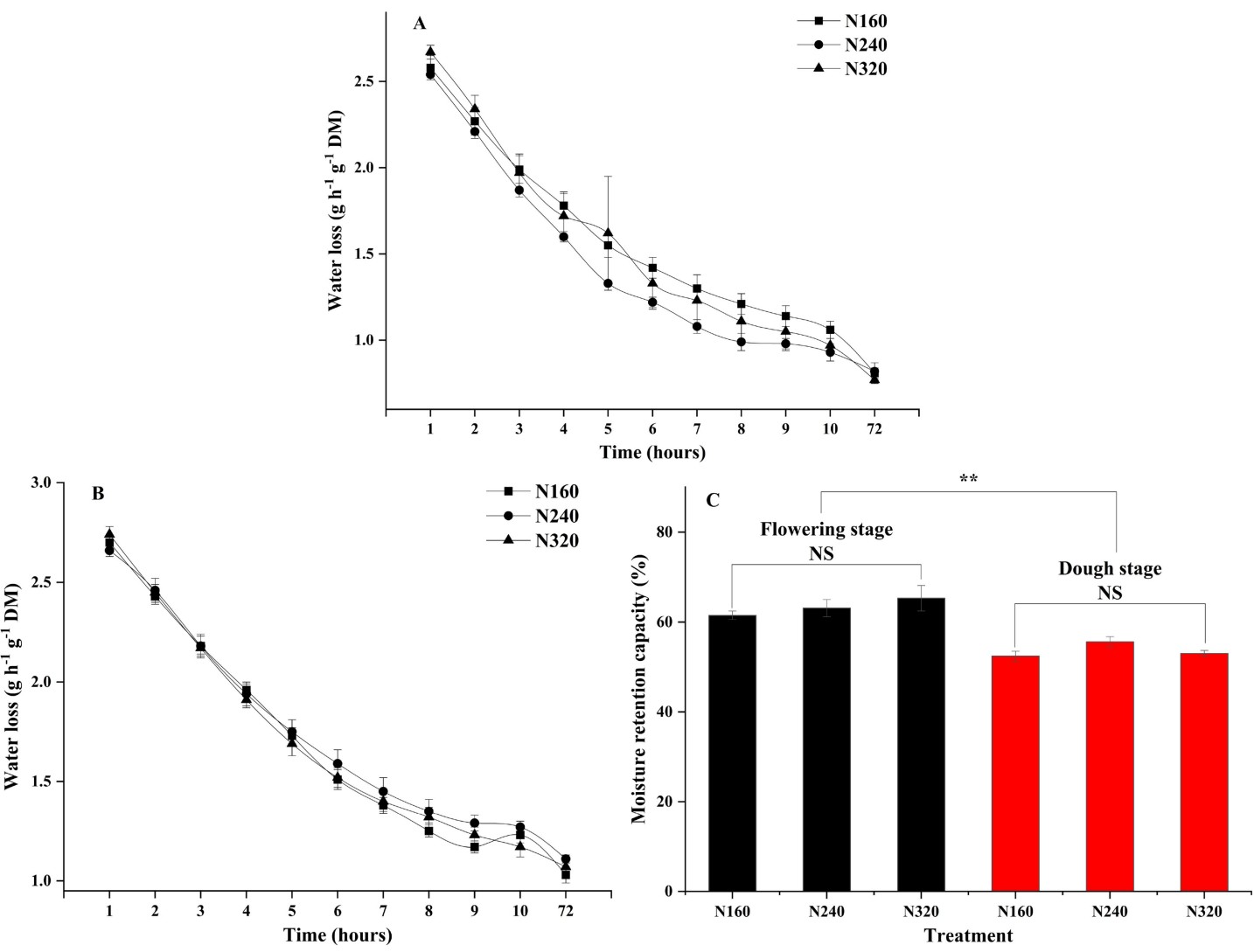

**Figure 2 Moisture retention capacity of leaves under different treatments and maturity stages.** Error bars represent standard errors. Black bars indicate the flowering stage and red bars indicate the dough stage. Asterisks indicate significant differences in moisture retention capacity at *P* < 0.01 (**) between the flowering stage and dough stage maturity stages; NS, not significant. (A) Flowering stage; (B) dough stage; (C) flowering stage and dough stage.

N320 treatments had a total of 114, 135, and 93 OTUs, respectively. Among these, 87 OTUs were shared between N160 and N240 samples, 75 OTUs were shared between N240 and N320 samples, and 71 OTUs were shared between N160 and N320 samples. The community abundance of *Alcaligenaceae*, *Erwiniaceae*, and *Pseudomonadaceae* families had higher abundance compared to other species (Fig. 6). At the genus level, the abundance percentages of *Bordetella*, *Pantoea*, and *Pseudomonas* were higher than shose of other species (Fig. 7). Notably, the percentage of *Pantoea* community abundance was significantly lower in the N320 treatment than those of the N160 and N240 treatments. The pH value, free amino acid concentration, and transpiration rate of leaves had a substantial effect on the abundance of microorganisms (Fig. 8). Specifically, the pH value

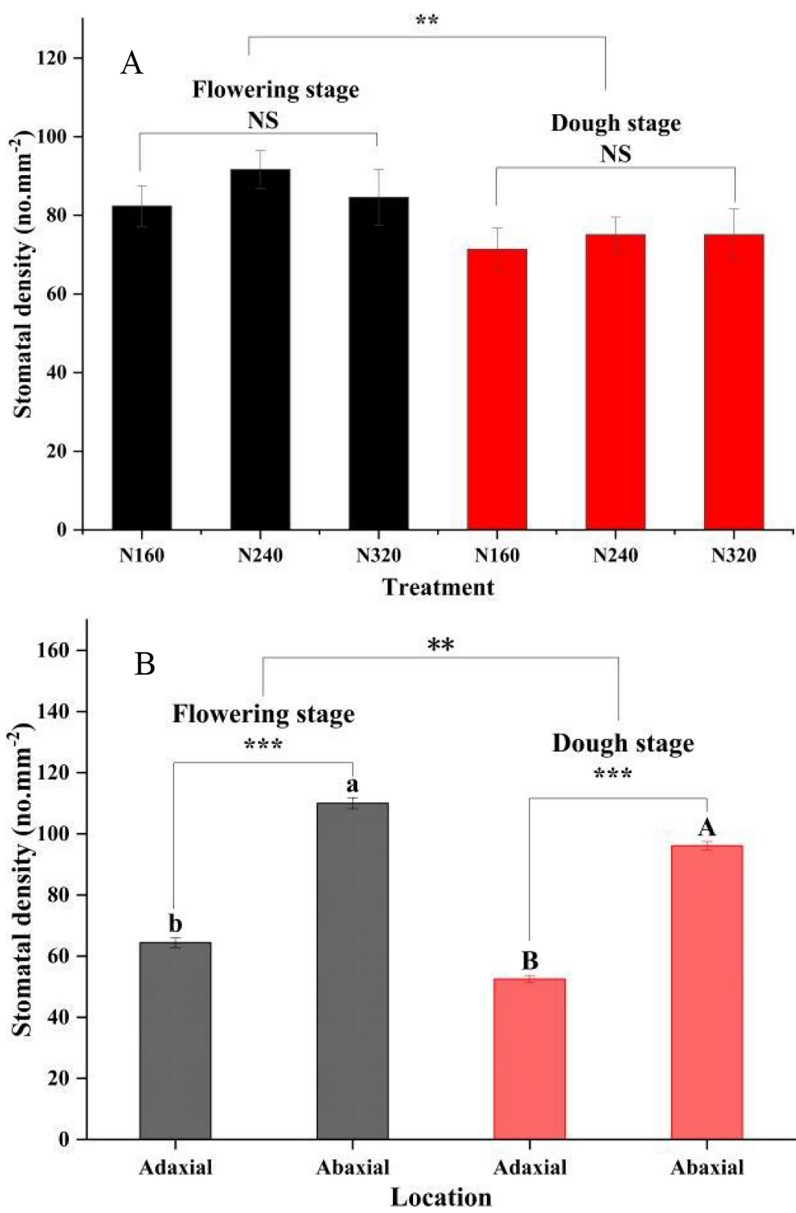

**Figure 3 Stomatal density of leaf for different treatments and maturity stage.** Error bars represent standard errors. Black bars indicate the flowering stage and red bars indicate the dough stage. Asterisks indicate significant differences in stomatal density at $P < 0.01$ (**), and $P < 0.001$ (***) between the flowering stage and dough stage maturity stages; NS, not significant. Different uppercase or lowercase letters at the top of the columns show significant differences on the adaxial or abaxial leaf surfaces among species, respectively. (A) N application rate; (B) Adaxial and Abaxial.

showed a positive correlation with the abundances of *Bordetella* and *Achromobacter* abundances ($P < 0.05$) (Fig. 9).

## DISCUSSION

Previous studies have indicated variations in both species and abundance of epiphytic microorganisms (*Kasozi, Kaiser & Wilhelmi, 2022*). The physiochemical properties of

**Table 4 Correlation coefficients between chemical and physical properties and population sizes of epiphytes ($n$ = 18).**

| Items | | Aerobic bacteria | Lactic acid bacteria | Molds | Yeasts |
|---|---|---|---|---|---|
| Chemical properties | Inorganic phosphorus | 0.672** | 0.732** | 0.688** | 0.752** |
| | Soluble sugar | 0.706** | 0.865** | 0.815** | 0.893** |
| | Soluble protein | 0.778** | 0.782** | 0.799** | 0.844** |
| | Free amino acids | 0.799** | 0.803** | 0.807** | 0.896** |
| | Total phenolic | 0.742** | 0.854** | 0.835** | 0.840** |
| | pH | 0.133 | 0.037 | −0.217 | 0.084 |
| Physical properties | Water loss | −0.500* | −0.667** | −0.497* | −0.617** |
| | Moisture retention capacity | 0.695** | 0.708** | 0.569* | 0.825** |
| | Stomatal density | 0.770** | 0.684** | 0.663** | 0.738** |
| | Transpiration rate | 0.349 | 0.503 | 0.125 | 0.566 |
| | Net photosynthetic rate | 0.019 | 0.525 | 0.253 | 0.319 |
| | Intercellular carbon dioxide concentration | 0.541 | 0.315 | 0.067 | 0.565 |
| | Intercellular carbon dioxide partial pressure | 0.541 | 0.315 | 0.067 | 0.565 |
| | Pore conductivity of water vapor | 0.393 | 0.526 | 0.018 | 0.633 |
| | Total conductance of water vapor | 0.392 | 0.523 | 0.023 | 0.627 |
| | Total conductivity of carbon dioxide | 0.392 | 0.523 | 0.023 | 0.628 |

**Note:**
Coefficients of non-parameter Spearman's correlation. Asterisks indicate significant differences at $P < 0.05$ (*) and $P < 0.01$ (**), respectively.

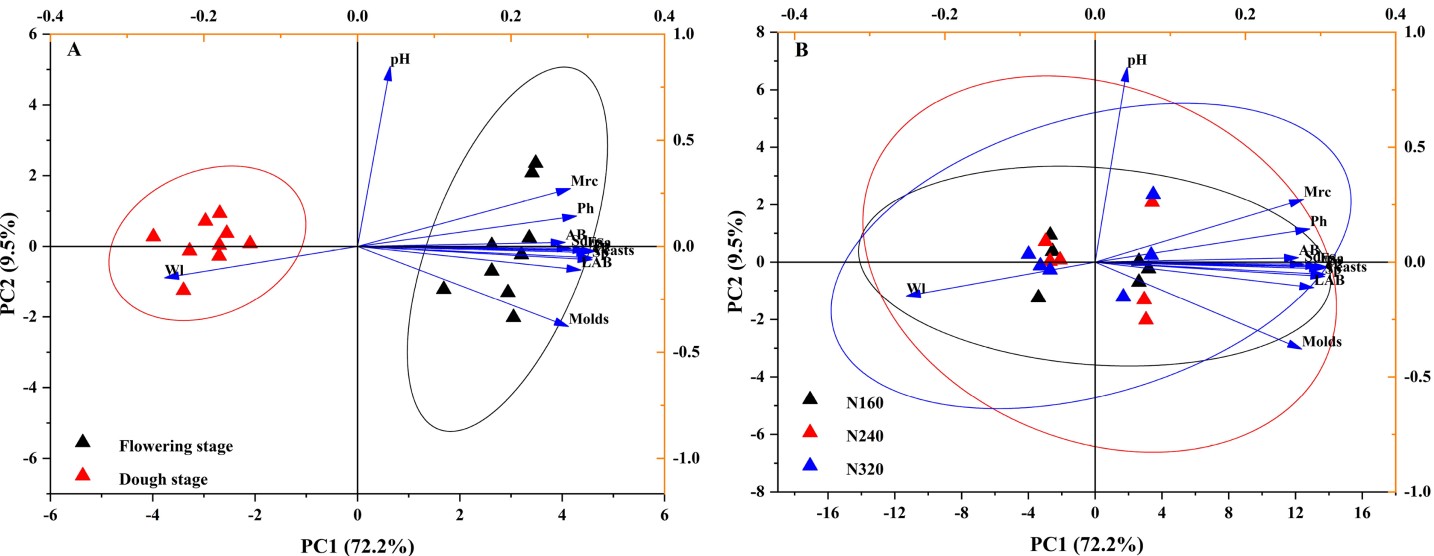

**Figure 4 Projection of different maturity stage (A) or N application rate (B) on the first two principal components based on their physicochemical characteristic attributes.** The first axis accounts for 72.2% of the total variance and the second for 9.5%. The original attributes, with their vectors intersecting at (0, 0), are also inserted. The length of each attribute vector is proportional to its contribution to the principal component axis. The ellipse indicates 95% confidence. AB, aerobic bacteria; LAB, lactic acid bacteria; Ph, inorganic phosphorus; Ss, soluble sugar; Sp, soluble protein; Faa, free amino acids; Tp, total phenolic; WI, water loss; Mrc, moisture retention capacity; Sd, stomatal density.

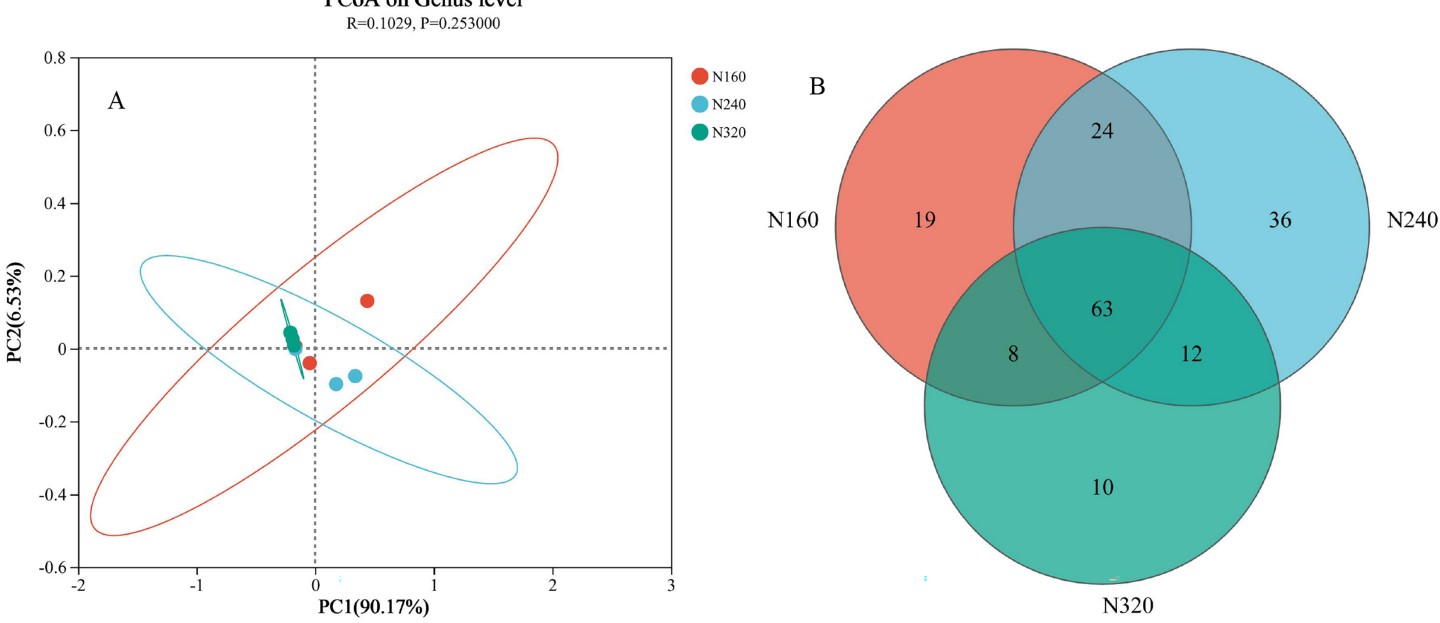

**Figure 5 Principal coordinate analysis (A) and venn diagram plot (B) of microbial composition at different N application rates.** The first axis accounts for 90.17% of the total variance and the second for 6.53%. The original attributes, with their vectors intersecting at (0, 0), are also inserted. The length of each attribute vector is proportional to its contribution to the principal component axis. The ellipse indicates 95% confidence. N160, red; N240, blue; N320, greener.

leaves can influence the abundance of epiphyte (*Tang et al., 2022*). In this study, the N application rates did not have a significant effect on the population sizes of microbial species on the surface of maize leaves (Table 1). This finding contradicts the results of a previous study (*Li et al., 2016*). The discrepancy might be attributed to the specific crop type or the N application rates. For instance, in a study involving Italian ryegrass, as the N application rate increased from 0 kg ha$^{-1}$ to 175 kg ha$^{-1}$, the LAB population initially increased and then decreased, but there were no significant changes in the populations of AB, yeasts, and molds (*Chen, Dong & Zhang, 2021*). Similarly, when studying wheat, increasing the N application rate from 0 kg ha$^{-1}$ to 300 kg ha$^{-1}$ resulted in a gradual decrease in the population size of AB, an initial increase followed by a decrease in LAB, an increase in yeasts, and a fluctuation in molds (*Li et al., 2016*). Generally, leaf surfaces exhibit lower nutrient levels, and the concentration of these nutrients is closely related to the concentration within the leaf's internal tissue (*Tang et al., 2023*). These nutrients, exuded from the leaf interior to the surface, play a crucial role in the colonization of microbial communities on the leaf surface. *Tian et al. (2020)* found that high N concentrations in leaves were associated with lower diversity of phyllosphere bacterial communities. Notably, the impact of N on microbial diversity is indirect. When plants are stimulated by N fertilization, changes in transmembrane nutrient transport of nutrients on the leaf surface, reservoir/source relationships, and water transport are observed (*Darlison et al., 2019*). These factors often determine the number and species of microorganisms present on the leaf surface.

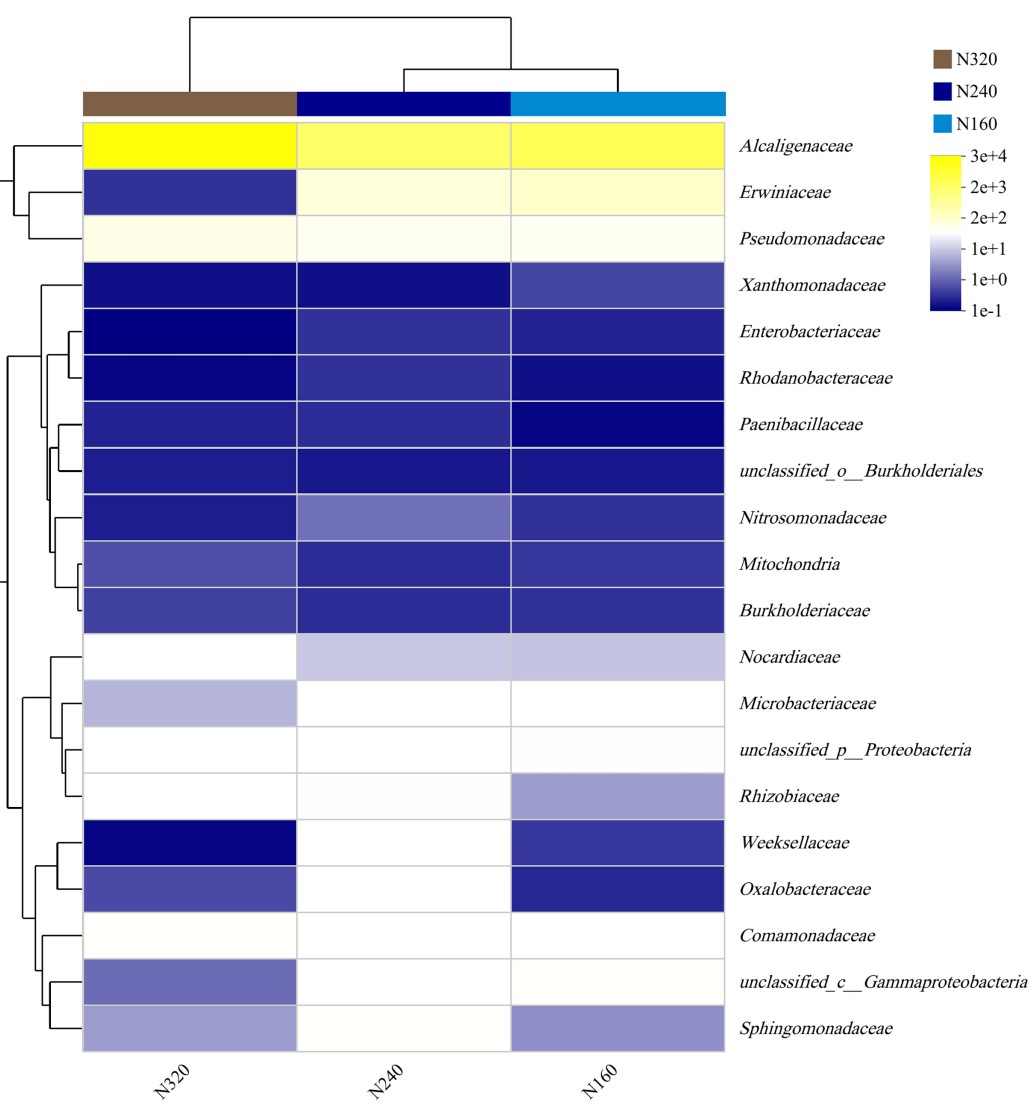

**Figure 6 Community heatmap of (20 most abundant) bacterial families on maize leaves under different N application rates.**

Protein metabolism is generally influenced by the rate of N supply, as increased N availability leads to elevated hydrolase activity and accumulation of soluble proteins in plants (*Liu et al., 2021*). Therefore, the N application rate is closely associated with soluble protein concentration. Interestingly, the present study found no significant effect of N application rates on soluble protein concentration, possibly because silage maize was used, resulting in a low total N per leaf (13 leaves plant$^{-1}$). Moreover, the relatively small differences in N fertilizer application rates might have contributed to these results. Apart from water, nutrient stress is a crucial factor influencing microbial survival and reproduction on the leaf surface. As the N supply rate increases, plants accumulate more soluble sugars and inorganic phosphorus (*Liu, Song & Liu, 2018*), regulating their osmotic and water-holding capacities. However, these nutrients must pass through stomata to be excreted onto the plant surface. In this study, the concentration of chemical substances on

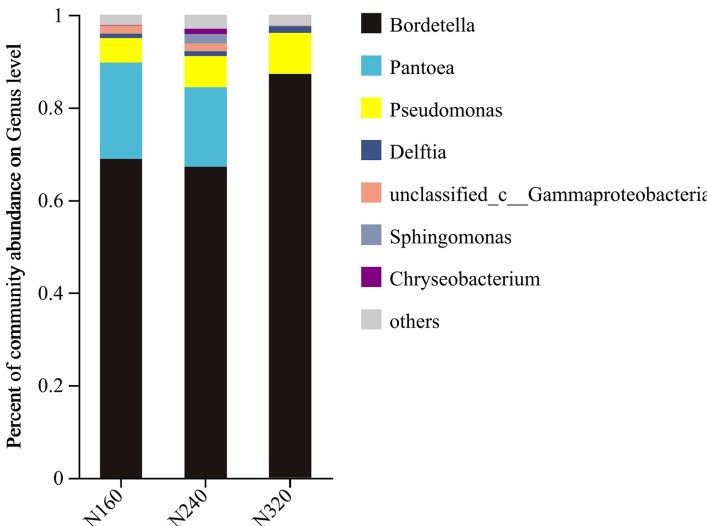

**Figure 7 Community barplot analysis of bacterial genera on maize leaves under different N application rates (genus level).**

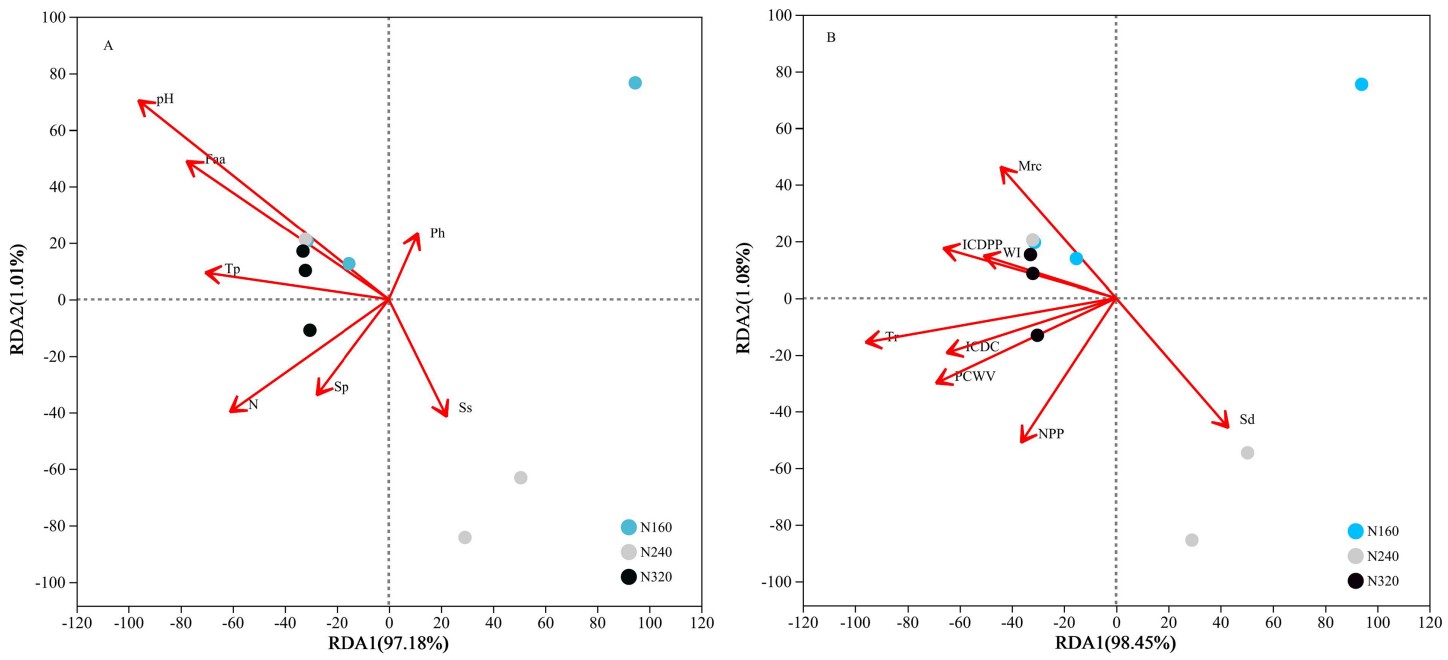

**Figure 8 Influence of leaf physicochemical properties on bacterial communities.** The first axis accounts for 97.18% (A: chemical properties) or 98.45% (B: physical properties) of the total variance and the second for 1.01% (A: chemical properties) or 1.08% (B: physical properties). The original attributes, with their vectors intersecting at (0, 0), are also inserted. The length of each attribute vector is proportional to its contribution to the principal component axis. N, N application rates; Ph, inorganic phosphorus; Ss, soluble sugar; Sp, soluble protein; Faa, free amino acids; Tp, total phenolic; WI, water loss; Mrc, moisture retention capacity; Sd, stomatal density; Tr, transpiration rate; NPP, net photosynthetic rate; ICDC, intercellular carbon dioxide concentration; ICDPP, intercellular carbon dioxide partial pressure; ICDPP, intercellular carbon dioxide partial pressure; PCWV, pore conductivity of water vapor. N160, red; N240, blue; N320, greener.

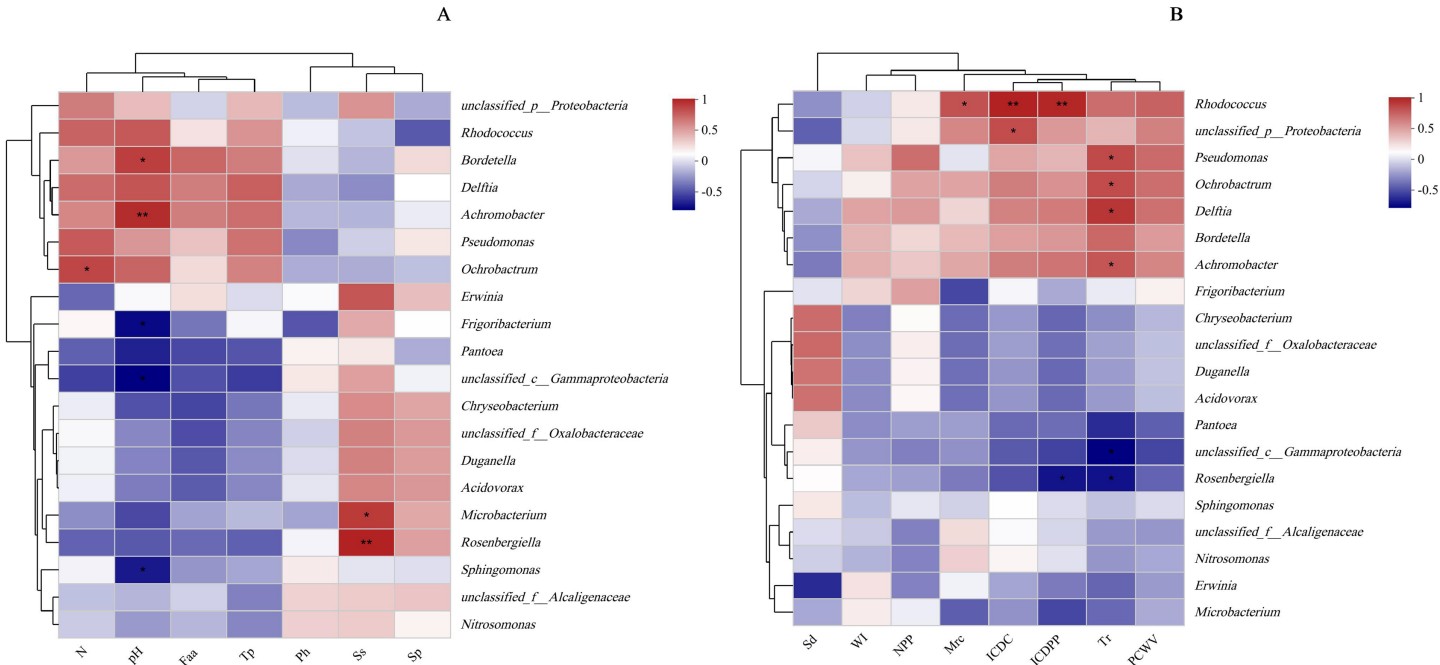

**Figure 9 Relationship between leaf physicochemical properties and bacterial communities.** The X-and Y-axes are environmental factors and species, respectively, and correlation R and *P* values are obtained by calculation. R values are presented in different colors in the figure, *P* values are marked with * if *P* < 0.05, and the legend on the right side shows the color intervals of different R values; asterisks indicate significant differences at *P* < 0.05 (*) and *P* < 0.01 (**), respectively. N, N application rates; Ph, inorganic phosphorus; Ss, soluble sugar; Sp, soluble protein; Faa, free amino acids; Tp, total phenolic; WI, water loss; Mrc, moisture retention capacity; Sd, stomatal density; Tr, transpiration rate; NPP, net photosynthetic rate; ICDC, intercellular carbon dioxide concentration; ICDPP, intercellular carbon dioxide partial pressure; ICDPP, intercellular carbon dioxide partial pressure; PCWV, pore conductivity of water vapor. (A) Chemical properties; (B) Physical properties.

maize leaf surfaces showed positive correlations with microbial population sizes, consistent with the findings of *Tang et al. (2022)*. Additionally, *Mercier & Lindow (2000)* concluded that the abundance of carbon sources, N sources, and inorganic molecules required by certain microorganisms on plant surfaces were closely related to microbial population sizes. There is evidence that yeast reduces the abundance of organic N on tall fescue (*Festuca arundinace*a Schreb.) leaves (*Nix-Stohr, Burpee & Buck, 2008*). Bacteria have also been shown to utilize sugars on leaf surfaces to increase their population size (*Chen, Dong & Zhang, 2021*). In the current study, *Microbacterium* and *Rosenbergiella* exhibited sensitivity to sugar (Fig. 9). Furthermore, the study demonstrated that only a very small population of LAB was attached to the leaf surface, which is not surprising considering that LAB are facultative anaerobic microorganisms and are therefore less abundant on leaf surfaces. The pH on leaf surfaces showed a positive relationship with the abundances of *Bordetella* and *Achromobacter*, likely because *Bordetella* and *Achromobacter* are basophilic.

Surprisingly, the present study found a positive relationship between total phenolic concentration and microbial population size, contrary to the results of *Yadav, Karamanoli & Vokou (2005)*. This discrepancy may be explained by certain microorganisms utilizing small molecule compounds. Metabolic activity in the dough stage was weaker compared to the flowering stage, resulting in fewer nutrients both inside and on the surface of leaves.

Reduced nutrient availability is not conducive to microbial colonization, explaining the simultaneous occurrence of smaller microbial populations and fewer nutrients on leaf surfaces during the dough stage.

High water content in leaves helps alleviates drastic temperature changes and keeps the leaf surfaces moist, promoting microbial reproduction. Furthermore, increasing environmental humidity raises the water activity on the plant surface, enhancing water and nutrient utilization by microorganisms (*Zi et al., 2021*). Improved leaf wetting properties facilitate the exudation of nutrients from the interior of leaves to the leaf surface (*Knoll & Schreiber, 2000*), providing more nutrients for surface microbial colonization. However, regardless of the abundance of leaf nutrients, they must pass through stomata to reach the leaf surface. Higher stomatal density increases leaf epidermal conductivity, facilitating microbial population growth on the leaf surface. In our study, the transpiration rate showed a positive correlation with the abundance of *Pseudomonas*, *Ochrobactrum*, *Delftia*, and *Achromobacte*r in the current study ($P < 0.05$), possibly because increased stomatal density promotes transpiration and enhances water exchange between plants and the air. Maintaining a consistently higher transpiration rate improves leaf wetting and promotes microbial colonization. This could explain the negative correlation observed in this study between the water loss rate from leaves and the population sizes of the four species, while moisture retention capacity and stomatal density exhibited positive correlations with these microbes (Table 4).

Currently, next -generation sequencing approaches are widely used for analyzing microbial species, effectively revealing the relationships between field management techniques and microorganisms (*Chen et al., 2018*). According to the PCoA results, the different sample points were closely clustered together (Fig. 5), indicating similar species composition at different N application rates. The abundances of bacteria did not significant increase with higher N application rates in this study (Figs. 6 and 7), suggesting similar species richness of maize leaves under different N application rates. This similarity is closely related to the nutrient deficiency on the surface of maize leaves (Table 2). The larger size and lower density of stomata further explain these findings. Generally, the length of abaxial stomata tends to increase while stomatal density tends to decrease with higher N application rates *(Cai et al., 2017)*. Lower stomatal density hinders the entry of $CO_2$ into the leaves and reduces plant water vapor conductance (*Lawson & Blatt, 2014*), resulting in inhibited water exudation from plants. As plants age, the stomatal density on leaves decreases, leading to reduced transpiration and a tendency of decreased water availability on the leaf surface. These factors contribute to the reduction in microbial species and populations. *Bordetella* (*Sukweenadhi et al., 2022*), *Pantoea*, and *Pseudomonas* (*Sun et al., 2021*) are common microorganisms found on maize, with the soil being a primary source. The present study found no evidence linking the abundance of these microorganisms to the N application rate. This finding explains the observed similarity in evolutionary relationships among microbial species under different N application rates in our study (Fig. 10). However, these do not hinder producers from using nitrogen fertilizer to regulate the number of lactic acid bacteria on the surface of forage, thereby improving

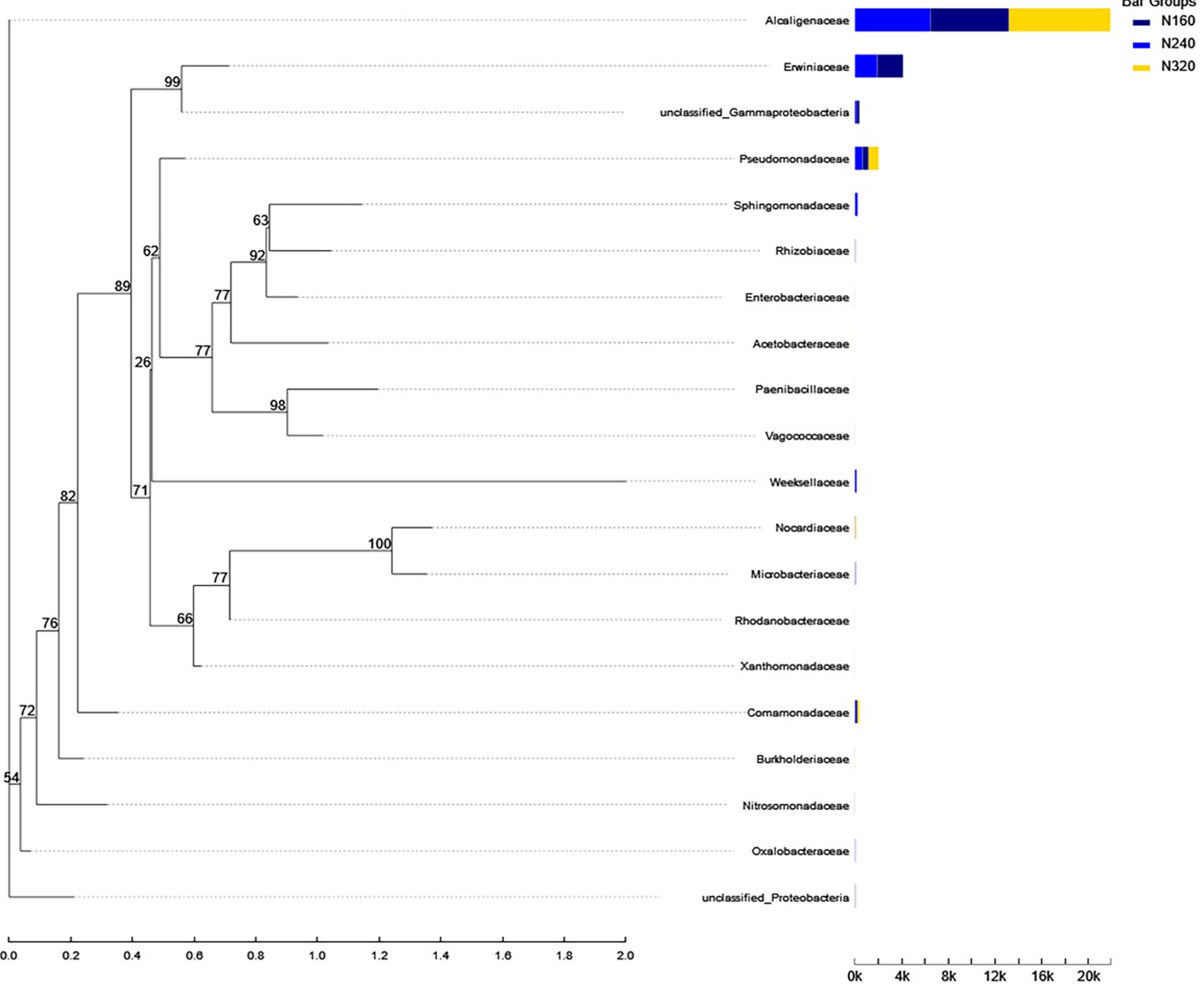

**Figure 10 Phylogenetic tree of microorganisms under different nitrogen application rates.** The phylogenetic evolutionary tree is shown on the left: each branch in the evolutionary tree represents a class of species, the branches are colored according to the advanced taxonomic level to which the species belongs, and the branch length is the evolutionary distance between two species, *i.e.*, the degree of difference of the species; the bar graph on the right shows the percentage of reads of the species in different groupings.

fermentation quality, as previous studies have confirmed the feasibility of this operation (*Li et al., 2016*).

## CONCLUSION

In conclusion, this study highlights the significant impact of maturity stage on microbial populations, physiological characteristics, and chemical compositions on maize leaf surfaces. The findings indicate that the abundance of nutrients and epiphyte populations is

higher during the flowering stage compared to the dough stage. Surprisingly, the N application rate did not have a substantial effect on microbial species or population size. Based on the results, we recommend applying approximately 160 kg ha$^{-1}$ of N fertilizer for optimal results in silage maize production. Additionally, we advise harvesting at the flowering stage is advised as it not only improves N fertilizer efficiency but also ensures favorable nutrient levels. These findings provide valuable insights for the management of harvest times and N fertilizer application in silage maize cultivation.

### Funding

This work was supported by the Yunnan Fundamental Research Projects (202301AU070035), the Xingzhao Talent Support Program, and the Scientific Research Fundamental Project of Yunnan Provincial Department of Education, China (2023J1205). The funders had no role in study design, data collection and analysis, decision to publish, or preparation of the manuscript.

### Grant Disclosures

The following grant information was disclosed by the authors:
Yunnan Fundamental Research Projects: 202301AU070035.
Xingzhao Talent Support Program.
Scientific Research Fundamental Project of Yunnan Provincial Department of Education, China: 2023J1205.

### Competing Interests

The authors declare that they have no competing interests.

### Author Contributions

- Dan Wu conceived and designed the experiments, analyzed the data, prepared figures and/or tables, authored or reviewed drafts of the article, and approved the final draft.
- Xueling Ma conceived and designed the experiments, analyzed the data, prepared figures and/or tables, authored or reviewed drafts of the article, and approved the final draft.
- Yuanyan Meng conceived and designed the experiments, prepared figures and/or tables, and approved the final draft.
- Rongjin Cai conceived and designed the experiments, prepared figures and/or tables, and approved the final draft.
- Xiaolong Zhang conceived and designed the experiments, prepared figures and/or tables, and approved the final draft.
- Li Liu conceived and designed the experiments, prepared figures and/or tables, and approved the final draft.
- Lianping Deng conceived and designed the experiments, prepared figures and/or tables, and approved the final draft.
- Changjing Chen conceived and designed the experiments, performed the experiments, prepared figures and/or tables, and approved the final draft.

- Fang Wang conceived and designed the experiments, performed the experiments, prepared figures and/or tables, and approved the final draft.
- Qingbiao Xu conceived and designed the experiments, performed the experiments, prepared figures and/or tables, and approved the final draft.
- Bin He performed the experiments, prepared figures and/or tables, and approved the final draft.
- Mingzhu He performed the experiments, prepared figures and/or tables, and approved the final draft.
- Rensheng Hu performed the experiments, prepared figures and/or tables, and approved the final draft.
- Jinjing Zheng performed the experiments, prepared figures and/or tables, and approved the final draft.
- Yan Han performed the experiments, prepared figures and/or tables, and approved the final draft.
- Shaoshen He performed the experiments, prepared figures and/or tables, and approved the final draft.
- Liuxing Xu conceived and designed the experiments, analyzed the data, prepared figures and/or tables, authored or reviewed drafts of the article, project leader, and approved the final draft.

## DNA Deposition

The following information was supplied regarding the deposition of DNA sequences:

https://www.majorbio.com/web/login/passport/login-email

## Data Availability

The raw data are available in the Supplemental Files.

The sequences are available at NCBI: SAMN36290997, PRJNA989725.

## Supplemental Information

Supplemental information for this article can be found online at http://dx.doi.org/10.7717/peerj.16386#supplemental-information.

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
