# Peer review of "Impact of nitrogen application and crop stage on epiphytic microbial communities on silage maize leaf surfaces"

_PeerJ, doi:10.7717/peerj.16386_

## Round 0.1 · original submission · Major Revisions

Check the manuscript thoroughly and improve it. Improve language as well.

Abstract
L20- delete N160, N240 and N320
L21- on ‘microbial diversity’ not ‘microbial species’
L22- replace compare with evaluated
L26- ‘total phenolic content’ not ‘total phenolic’

Introduction
Add a couple of sentences on flowering and dough stage.
L47- microorganisms not microorganism
L52- correct “correlated with leaf morphological characteristics of the leaves” (leaf two times)
L72- increase “microbial diversity” not “microbial species diversity”
L72- and their abundance
L74- “microbial diversity” not “microbial species diversity”

Methodology
Determination of microorganism
L110- add “and” and correctly write De Man, Rogosa and Sharpe agar medium and also mention its well-known abbreviation (MRS media)
Measurement of Chemical Constituents
L115- “To extract chemicals from the leaf surface 20 g of fresh material were washed” describe which material
Measurements of Physiological Characteristics
L113- air flow rate not “flow rate”
Microbial identification
L141- clear extra space in this line
L144- remove the word “PCR product” from end of sentence
L148- Raw sequencing data not raw sequenced sequences
L149- chimeric reads not chimeras
“using the thermocycler PCR system (GeneAmp 9700, ABI, USA). PCR products from the same sample were mixed and recovered using 2% agarose gel PCR products. An AxyPrep 144 DNA Gel Extraction Kit (Axygen Biosciences, Union City, CA, USA) was used to purify recovered 145 products and 2% agarose gel electrophoresis was conducted” rephrase and don’t write agarose gel electrophoresis two times
Data Processing and Statistical Analysis
L154-Between bacterial diversity not the number of microbial colonies
L156- “to clarify the relationships 15 between physiochemical characteristics and microbial population sizes of the different leaves.” Rephrase the sentence and remove grammatical mistakes.

Results
Epiphytic Population Relationships with Physicochemical Traits of Leaf
L200-203 recheck the statement technically and correct it.
Microbial Composition of Leaf
L212- and figure no 10: correct the interpretation (it does not exceed 60%, you are confusing Bordetella with others). Correct color for Bordetella and others in figure.

Discussion
L282-Remove space from next-generation
L289- “while stomatal density tends to decrease with higher N application rates” write in regular ( un-italic) writing
Discuss the importance of 16S r RNA in microbial diversity
Table 4 title, correct ‘and and’

**Language Note:** The Academic Editor has identified that the English language must be improved. PeerJ can provide language editing services - please contact us at copyediting@peerj.com for pricing (be sure to provide your manuscript number and title). Alternatively, you should make your own arrangements to improve the language quality and provide details in your response letter. – PeerJ Staff

·

Basic reporting

I would suggest improvements in English. Overall paper is well written. Add some more references in introduction as they are not sufficient.

Experimental design

I am satisfied with the experimental design. Kindly add picture of the experiment as well.

Validity of the findings

Satisfied.

Additional comments

Good effort!
Overall paper is good. Kindly make changes as I suggested.

·

Basic reporting

The manuscript demonstrates clear, unambiguous, and professional English language usage throughout. The introduction and background provide a comprehensive context for the study. The literature is well-referenced and relevant to the research topic. The structure adheres to PeerJ standards, discipline norms, and has been improved for clarity. The figures are relevant, of high quality, well-labelled, and thoroughly described. Furthermore, the authors have supplied the raw data in accordance with PeerJ policy.

Experimental design

The original primary research falls well within the scope of the journal. The research question is clearly defined, relevant, and carries meaningful implications. The study explicitly highlights how it addresses an identified knowledge gap in the field. A rigorous investigation has been conducted, meeting high technical and ethical standards. It would be beneficial to include details about crop husbandry in the Materials and Methods section, such as the application method of fertilizers, weed control measures, and the quality of irrigation water.

Validity of the findings

The findings of the research are valid and well presented.
Suggestions to improve the manuscript

It would be beneficial to include details about crop husbandry in the Materials and Methods section, such as the application method of fertilizers, weed control measures, and the quality of irrigation water.

In the Discussion section, it would be helpful to add some introductory sentences to provide context and guide readers into the key points of the discussion.

Furthermore, it is essential to carefully cross-check the references in both the manuscript text and bibliography to ensure accuracy, and make any necessary insertions or deletions.

Additional comments

Line # 19: Please write the full form of "N" in its first mention.
Line # 65: "N in the soil is very efficient" requires elaboration.
Line # 83-84: The average rainfall and temperature of the past 20 years are given; however, it is important to consider these parameters for the cropping season.
Line # 97 & 109: The heading is unclear; please make it more explicit.
Line # 190: “The fastest rate of water loss occurred before 8 h”. Please rephrase the sentence for better clarity.
Line # 233 & 249-250: Avoid repetitive use of the authors' names.
Line # 405: The reference is not cited in the text.

Reviewer 3 ·

Basic reporting

Overall, the study provides valuable insights into the dynamics of microbial populations on maize leaf surfaces and their relationship with N application rates. Addressing the following points will help enhance the manuscript impact and contribute to the scientific discourse in the field of plant-microbe interactions in agricultural systems.
Point 1: The introduction provides a solid overview of the research topic but lacks a clear research hypothesis or objective. Consider explicitly stating the research question you aimed to answer in this study.
Point 2: The methodology section is concise, but some details are missing. Please provide more information about the experimental design, such as the number of replicates and how treatments were randomized. Additionally, explain the statistical methods used for data analysis in more detail.
Point 3: While the results section presents the data clearly, further elaboration on unexpected findings is necessary. For instance, the positive relationship between total phenolic concentration and microbial population size contradicts a prior study. Provide possible explanations for this discrepancy.
Point 4: The discussion effectively compares your findings with previous research. To enhance clarity, consider starting each paragraph with a topic sentence that summarizes the main point of that paragraph.
Point 5: Expand on the practical implications of your results. How can farmers use this information to optimize N fertilizer application and improve crop management? Adding concrete examples will make the implications more tangible.
Point 6: Proofread the manuscript for grammar and clarity. Well-written prose will contribute to the overall readability and impact of the paper.

Experimental design

The experimental design is generally well-described, providing necessary details on the study location, crop species, and treatments. However, some key aspects need further clarification for a complete understanding of the study.
Point 1: It's important to explicitly mention the number of replicates for each treatment group. Providing this information will help assess the robustness of your results and conclusions. Additionally, if any statistical methods were used to determine the appropriate sample size, mention those methods and their rationale.
Point 2: Clarify the process of randomization of treatments and control groups. Detailing how the random assignment was conducted ensures that the treatment groups are comparable and reduces potential biases.
Point 3: While the N application rates are well-defined, specify the exact timing of N fertilizer application in relation to the growth stages of the maize plants. This will help readers understand how nutrient availability varied during different stages of plant growth.
Point 4: Provide information about the environmental conditions during the experiment. This could include factors such as temperature, humidity, and light conditions, which might influence microbial populations and physiological traits of the plants.
Point 5: Detail the methods used to measure physiological characteristics, chemical compositions, and microbial populations. Specify the equipment and protocols used for each measurement.

Validity of the findings

Point 1: The data presented in the manuscript seem reliable and well-documented. However, to enhance the validity of your findings, it's important to provide details on data collection procedures, including standard operating procedures for measurements, calibration methods for instruments, and any quality control measures taken during data collection.
Point 2: Address the representativeness of the samples used in your study. Clarify whether the chosen samples adequately reflect the larger population of interest (maize plants under similar conditions). Providing information on the selection criteria and potential biases in sample collection will strengthen the validity of your results.
Point 3: Discuss how your findings align with or deviate from previous research in similar contexts. Address any discrepancies and provide plausible explanations for differences in results.
Point 4: Consider the long-term implications of your findings. How might the observed effects on microbial populations, physiological traits, and chemical compositions impact the overall health and productivity of maize plants over multiple growing seasons? Discuss potential sustainability and practical implications.

Additional comments

Overall, the manuscript effectively presents the research conducted on the effects of nitrogen application rates and maturity stages on microbial populations and physiological traits of maize leaves. Addressing the following points will further enhance the clarity, coherence, and impact of your manuscript
General comments:
1. Title should be revised as "Impact of Nitrogen Application and Crop Stage on Epiphytic Microbial Communities on Silage Maize Leaf Surfaces"
2. In this manuscript, some abbreviations are introduced without their full form in the text. For clarity, consider providing the full form of an abbreviation the first time it appears, followed by the abbreviation in parentheses. For example, "Leaf Area Index (LAI)".
3. Ensure that all references cited in the text are included in the reference list and vice versa. Use a consistent citation style throughout the manuscript.
4. I suggest combining some of the relevant figures. This approach will not only help reduce the overall figure count to 8, but also enhance the reader's understanding by presenting related information in a more cohesive manner. Below, I provide recommendations for combining certain figures:
Figures 2 and 3 both provide information about moisture retention capacity and stomatal density at different maturity stages. Consider combining these figures into a single figure to visually depict how these two physiological characteristics change over the course of the plant growth.
Figures 4 and 5 both discuss water loss rates and their relationship with microbial populations. Combining these figures can show how water loss rates vary with maturity stages and their corresponding impact on microbial abundance.
Figures 7 and 8 both pertain to the microbial composition of leaf surfaces and how it relates to N application rates. Combining these figures can create a comprehensive illustration of how microbial communities respond to different nitrogen levels.

---

## Round 0.2 · accepted · Accept

Authors have revised the manuscript according to the suggestions of reviewers and the current version is suitable for publication.